# Infant and Young Child Feeding Knowledge among Caregivers of Children Aged between 0 and 24 Months in Seshego Township, Limpopo Province, South Africa

**DOI:** 10.3390/healthcare11071044

**Published:** 2023-04-05

**Authors:** Ndivhudzannyi Muleka, Baatseba Maanaso, Mafiwa Phoku, Mabitsela Hezekiel Mphasha, Maishataba Makwela

**Affiliations:** Department of Human Nutrition and Dietetics, University of Limpopo, Polokwane 0790, South Africa

**Keywords:** infant and young child feeding, children under 24 months, breastfeeding, complementary feeding, knowledge

## Abstract

Background: Appropriate infant and young child feeding (IYCF) involves the initiation of breastfeeding within an hour of delivery, exclusive breastfeeding for 6 months, introduction of complementary feeding at 6 months while continuing breastfeeding for 2 years or beyond. Adequate IYCF knowledge among caregivers is associated with improved practices, lowers risk of kids developing malnutrition, infection, morbidity, and mortality. Early introduction of solid foods, mixed feeding, inadequate breastfeeding, and complementary feeding are all prevalent in South Africa. These are related to caregivers’ lack of IYCF knowledge. Hence, this study aims to determine the IYCF knowledge level of caregivers of children under 24 months in the semiurban Seshego Township, South Africa. Methodology: Quantitative and cross-sectional design was applied. A total of 86 caregivers were selected using simple random sampling, which is representative of a target population of 110. Structured questionnaire was utilised to gather data, and analysed through statistical software, using descriptive and inferential statistics. Chi-square test was used to calculate associations at 95% confidence interval, where a *p*-value of < 0.05 was considered statistically significant. Results: Findings show that 67% of participants had good IYCF knowledge (a score of 81 to 100%) and there was a significant relationship between knowledge and education (*p* = 0.001). Moreover, 40.7% did not know that exclusive breastfeeding should be up to 6 months, and 90% mentioned that breastmilk protects the child against diseases. Most participants (82.6%) know that complementary feeding should be introduced at 6 months with continuation of breastfeeding. Conclusions: Caregivers know that breastfeeding should begin immediately after birth, and that it protects against diseases. Moreover, they know that solid food should be introduced at 6 months. However, there is still a need to strengthen IYCF education, particularly on exclusive breastfeeding. Interventions to improve IYCF knowledge should be intertwined with improving educational and health literacy on breastfeeding and complementary feeding.

## 1. Background

Infants and young children need to be properly fed in order to maximise growth and development throughout their first 1000 days of life and boost their chances of survival [1,2]. For infants and young children to grow, stay healthy, and develop to their full potential, they must receive adequate nourishment. Inadequate nutrition increases the risk of sickness, long-term growth, and health impairments [1]. It may also lead to childhood obesity, which is a major public health problem.

Breast milk is a baby’s first natural food source and it provides all the nutrients and energy they need for their first six months of life [2]. It provides newborns with the nourishment they need, aids in the development of their immune systems, and has the ideal balance of fat, sugar, water, protein, and vitamins [3]. Breast milk protects infants from viral and chronic illnesses while fostering sensory and cognitive development [4]. Moreover, breastfeeding promotes a balanced weight range, which lowers the incidence of childhood obesity [5]. An increase in breastfeeding rates could save roughly 823,000 children under the age of five each year over the world [6]. The risk of death from pneumonia, diarrhoea, and newborn sepsis is decreased with the promotion, support, and protection of breastfeeding [7]. Moreover, it can greatly reduce mortality and morbidity due to all conditions caused by general infections, including gastrointestinal and respiratory tract infections [8,9]. Appropriate breastfeeding involves on-demand feeding, which can take away a mother’s independence from child-feeding duties [10], and limits her time to engage in other useful activities, such as work. Exclusive breastfeeding practiced during the first six months of life improves cognitive function and academic achievement, and may also help in minimising risk of obesity and chronic diseases, such as cancer and cardiovascular disease, later in life [11,12]. Several studies have shown that adhering to the EBF can reduce the risk of infant deaths due to pneumonia, which may further reduce the mortality rate of children below five years of age [13,14]. Breast milk falls short of infants’ nutritional needs as they become more active, particularly from the age of six months; this gap widens as babies and young children get older [15].

Complementary feeding is crucial for bridging these gaps. Good complementary feeding is defined as having a varied diet, eating frequently, and introducing extra food on schedule [16]. Children should be aggressively encouraged to eat by their caregivers, who should also be alert to the child’s hunger cues. Sufficient and safe complementary feeding, supplied timely and correctly, is required. Timely in the sense that it should be introduced when the infant’s demand for nutrients and energy exceeds the amount that can be supplied by breastfeeding alone. Appropriate nutrition refers to providing infants with an appropriate amount of energy, protein, and micronutrients to support their growth. Safe foods are those that have been hygienically prepared, stored, and consumed with clean hands and utensils. Providing food in accordance with a child’s hunger and fullness cues and following age-appropriate meal and feeding schedules constitutes being well fed. In order to provide a diverse nutrient intake that satisfies the developing infant’s requirements for all nutrients, the WHO advises diet diversification in addition to timely introduction. This means that as a part of the complementary feeding, a variety of the basic food groups should be offered [17]. These complementary foods should be given between the ages of 6 and 8 months at first 2–3 times per day, progressing to 3–4 times per day between the ages of 9 and 11 months, and thereafter 1–2 times per day as desired [17].

Malnutrition, vitamin deficiencies, and delayed motor and cognitive development in infants and young children are all health problems that can be further worsened by poor complementary feeding methods, which can also impede a child’s growth [18]. Inadequate breastfeeding and delayed introduction of complementary foods, increase the risk of childhood mortality and morbidity [19,20]. To prevent child malnutrition and death, breastfeeding must begin within an hour of delivery, be exclusive for the first six months with the introduction of timely and adequate quality complementary food, and last for at least two years [21]. Stunting is a condition that results from malnutrition in the first two years of life that leads an adult to be several centimetres lower than their optimal height [22]. There is proof that people who were undernourished as children perform less intelligently [23]. Moreover, they may be less capable of performing hard labour [24]. Women’s ability to reproduce is impacted by their childhood nutrition, and their babies may be born with lower birth weights and require more complex births [25]. However, only 41% of infants under the age of six months in the world receive breast milk, whilst 45% continue to breastfeed up to two years [7]. Research conducted in the Limpopo Province, South Africa, indicated that caregivers gave solid foods to infants before they were 6 months old. It was also discovered that exclusive breastfeeding was lower than the World Health Organisation (WHO) standards [26,27]. In many countries, only a few children obtain complementary foods that are safe and nutritionally adequate; just a minority of infants, aged between 6 and 23 months, satisfy the WHO standards for age-appropriate dietary diversity and feeding frequency [28]. According to Motebejana et al. [29] the amount of nutritional knowledge of caregivers can be used to predict their feeding practices, with caregivers’ lack of child nutritional knowledge having a substantial impact on children’s feeding habits.

A Ghanaian study found a link between nutritional awareness of caregivers and kids’ nutrition and health outcomes [30]. According to a survey conducted in Western Ethiopia, 93.8% of moms are aware of breastfeeding in general but not exclusively [2]. Most mothers did not feed their children according to the WHO recommendations [31]. High rates of stunting, poor feeding practices, low scores for dietary diversity, caloried intake, and nutrients intake point to the need to enhance/intensify nutrition education and strategies for complementary feeding [32]. Malise et al. [27] found that 75% of caregivers lacked knowledge regarding breastfeeding exclusively for a child’s first six months of life and that more than 40% of caregivers were unaware that an HIV-positive mother can exclusively breastfeed [2]. According to studies, between 48.6% and 90% of caregivers in underdeveloped nations, such as sub-Saharan Africa, are aware of the benefits of breastmilk to the children [28,33]. Between the ages of 6 and 36 months, height-for-age z-scores (HAZ) for children revealed a significant relationship to maternal childcare knowledge, with mothers who were less knowledgeable about nutrition having difficulty implementing the right complementary feeding, which resulted in low HAZ. Furthermore, it has been shown that parental and caregiver knowledge of nutrition protects kids from circumstances that result in subpar HAZ and weight-for-age (WAZ) z-scores [34,35]. Maternal knowledge of food ingredients and children’s and adolescents’ HAZ were found to be significantly correlated with maternal and child nutrition outcomes in urban Kenya [36]. The same study discovered compelling evidence connecting caregivers awareness to children’s HAZ and the negative consequences of dietary recommendations on health [36]. Educated/informed women were 5.5 times more likely to practice effective breastfeeding than caregivers with minimal knowledge of nutrition, according to an Ethiopian study [37].

In an effort to help and promote effective breastfeeding and complementary feeding, policies that supported and encouraged breastfeeding and complementary feeding were introduced and amended in South Africa [38]. According to Wu Q et al. [39] 85% of children get complementary feeding before the age of six months, and caregivers frequently provided solid foods either too early or too late [40]. Despite the fact that there are numerous possible causes for this, it has consistently been realised that practices and knowledge are interconnected. Therefore, less knowledgeable caregivers are likely to poorly or inadequately feed their infant compared to the knowledgeable caregivers. There is also a problem of improper feeding practices, such as early introduction of solid food, and mixed feeding. In addition, the increasing levels of malnutrition, which is also prevalent in Limpopo province among infants, therefore raises concerns regarding levels of knowledge of infant and young child feeding among caregivers. Hence, it is critical to have a baseline understanding of IYCF among caregivers in this area. The findings of this study, in future, will be useful in developing an intervention to address knowledge deficits within the overall strategy to curb malnutrition in the area. Since South Africa has adopted the WHO recommendations on infant feeding, this study is based on these recommendations. Therefore, the goal of this study is to determine the level of feeding knowledge among caregivers of 0- to 24-month-old infants who attend postnatal care in Seshego Zone 4 clinic, Limpopo province.

## 2. Methodology

### 2.1. Research Design and Setting

A cross-sectional survey and quantitative study approach was adopted for this research. The study was conducted at Seshego Zone 4 clinic, which is located in the township of Seshego within the Polokwane municipality of Limpopo Province, South Africa. The clinic offers antenatal and postnatal care to the Seshego locals as well as the general public, which is dominated by people speaking Sepedi. Caregivers are provided with infants and young children feeding education which is imparted by healthcare professionals. Sepedi is the language spoken by the majority of people in Seshego.

### 2.2. Population and Sampling of the Study 

The target population were caregivers of infants aged between 0 and 24 months who received primary health care services at Seshego Zone 4 clinic. As per the register of the clinic, there were 110 children aged 0–24, during the study period which was conducted between October and November 2022. Therefore, a total of 110 constitute the population size. In this context, the term “caregivers” refers to mothers and/or legal guardians who take on the primary duty of caring for children of the age 0–24 months. Only caregivers who could speak Sepedi and English and were at least 18 years old were included in the survey because they could provide informed consent. The sample size of 86 was calculated using the sample calculation formula developed by Krejcie and Morgan [41]. The formula is as follows: $S=\frac{x^{2}NP\left(1-P\right)}{d^{2}\left(N-1\right)+x^{2}P\left(1-P\right)}$ where *S* = required sample size; *X*^2^ = the table value of Chi-square for 1 degree of freedom at the desired level (3.841); *N* = The population size; *P* = the population proportion (assumed to be 0.50 since this would provide the maximum sample size); and *d* = the degree of accuracy expressed as a proportion (0.05).

### 2.3. Instruments and Data Collection

Data were gathered using a closed-ended questionnaire with two sections: demographic profile and knowledge. The demographic profile had twelve questions related to age, gender, income, education, relationship to the child, and age of the child. The knowledge section had six questions which included when should breastfeeding begin, period of exclusive breastfeeding, benefits of breastfeeding, and when should complementary feeding be introduced. The knowledge section used a three-point Likert scale (yes; no; and not sure), and caregivers had to respond accordingly. The questionnaire was developed keeping in mind the WHO recommendations [1,2]. Furthermore, the questionnaire was piloted to test for reliability. A pilot study was carried at the same clinic with participants who were not in main study. The content validity of the instrument was assured by comments provided by the two supervisors of this study and three dietitians working in a nearby tertiary hospital. The questionnaire was available in both Sepedi, the primary language spoken in Seshego, and English. All consenting caregivers received the questionnaires to fill out on their own. The questionnaires were filled out in front of the researchers responsible for data collection, who provided clarity and assistance where required.

### 2.4. Data Analysis

The Statistical Package for Social Sciences (IBM, London), software version 27, was used for data analysis. When calculating frequency distributions, means, and standard deviations, descriptive statistics were utilised. Using a Chi-squared test, relationships were calculated with a 95% confidence level. A *p*-value of 0.05 was considered to be statistically significant. Participants’ replies were further categorised into right and wrong solutions. The ‘not sure’ answers were deemed incorrect. Moreover, each response was worth one point, and all responses were further converted into percentage of 100. For the purposes of this study, knowledge was graded on 100% score and on a scale from poor to good. Poor knowledge is defined as receiving a total score between 0 and 50%, fair knowledge between 51 and 80%, and good knowledge between 81 and 100%.

### 2.5. Ethical Issues

The study was approved by the Turfloop Research Ethical Committee (TREC) at The University of Limpopo, which issued clearance certificate number TREC/488/2022: UG. However, this article is focused on feeding knowledge of caregivers of children between 0 and 24 months. All the study subjects signed a written informed consent form. Participants were informed that their freedom to leave the study at any moment without consequence was entirely voluntary. Participants’ personal information was also kept secret and confidential.

## 3. Results

### Socio-Demographic Profile 

Table 1 shows that most participant caregivers who brought their children to the clinic were of the age group 18–35 years (80.2%), had a secondary education or less (55.8%), unemployed (68.6%), and dependent on social grant (67.4%). Furthermore, majority were mothers of the children (97.7%), mostly had two children or less (60.5%), and 52.3% of the children were aged 0–6 months.

Table 2 shows that 96.5% of caregivers know that breastfeeding should be initiated immediately after birth, 40.7% did not know that exclusive breastfeeding must be provided up to 6 months, 90% alluded that breastmilk protects the child against diseases, and 27.9% did not know that the HIV exposed mothers should breastfed. Additionally, 82.6% know that complementary feeding should be introduced at 6 months and 82.6% said breastfeeding should not be stopped when complementary food is given.

According to Figure 1, 67% caregivers had good feeding knowledge, whereas 3.5% 28.7% caregivers had fair and poor infant and young child feeding knowledge, respectively.

Table 3 shows that there is no statistically significant difference between knowledge and age (*p* = 0.751), employment (*p* = 0.564), and number of children (*p* = 0.436). However, there was a significant relationship between the knowledge and education (*p* = 0.001) of participants.

## 4. Discussions

This study found that 67% of subjects have a sufficient understanding of feeding infants and young children. The findings were in line with cross-sectional studies carried out in Burkina Faso [23] and Ethiopia [2], which also showed that caregivers generally had good knowledge of infant and young child feeding (IYCF). However, in contrast, a South African study carried out in the Sekhukhune district of the province of Limpopo, indicated that the IYCF guidelines were not sufficiently understood [11]. Kenyan and Chinese studies [20,24] also indicated that caregivers lacked sufficient comprehension of the IYCF guidelines. In this study too, a sizable percentage of survey respondents (28.7%), showed inadequate comprehension, which calls for reinforcing of IYCF education. People frequently misinterpret pertinent health information given to them by healthcare professionals, which can result in dangerous mistakes while following advised care programmes [42]. Even if they may not fully understand the pertinent facts, patients may believe they understand the health information given to them. Patients may also make deceptive claims that they comprehend medical advice in an effort to keep a straight face, project competence, and convey control over their care [43,44]. People could feel ashamed to admit that they do not grasp important medical information or are puzzled about it. This may cause healthcare professionals to overestimate how well patients comprehend the health information that has been given to them [45]. Knowledge is an important predictor of how kids are fed. On the other hand, education might affect how well individuals comprehend other parts of the IYCF message. This study showed a strong link between education level and feeding knowledge, with a *p* value of 0.001. The findings are in line with a study by Williams et al. [46] conducted in Australia, which discovered a strong relationship between education level and nutritional knowledge. This study further showed that children of caregivers who had completed secondary school or higher are more likely to have a greater understanding of nutrition. Nutritional knowledge among the caregivers and inadequate maternal education have both been connected to children’s unhealthy eating habits [47]. Encouraging education should be at the centre of promoting nutritional infant and young child feeding knowledge. The effective delivery of healthcare depends on the patient’s understanding of treatment suggestions. The type and quality of the food consumed depends on the caregivers’ knowledge of nutrition, among other factors [48]. In addition to good attitudes and knowledge, as well as improved skills and more self-advocacy, education can lead to better lifestyle options. Patients, even those with advanced language abilities, sometimes struggle to comprehend medical information because they lack familiarity with medical jargon, are distracted with their symptoms, and find it difficult to concentrate while agitated [42,45]. Higher levels of education are also associated with good health, while lower educational levels are associated with illness, mortality, and disability [49]. While people of all literacy levels struggle to comprehend and apply health information, those with low literacy levels require extra support. They require assistance in reading the written material, and because they rely more on verbal explanations, they also require assistance in recalling what they heard [50]. In this context, lower educational levels could lead to poor breastfeeding, exclusive breastfeeding and complementary practices, which predisposes children to many health problems. Therefore, this study recommends an effective strategy, which includes using simple or home language and peers with adequate knowledge to improve the child-feeding knowledge of caregivers with low educational levels.

In this study, only 59.3% of participants said that it was good to breastfeed exclusively for the first six months. Findings of this study are in line with that of Frans et al. [51] who discovered that 56.6% of women were aware of the exclusive breastfeeding period of six months. In contrast, a study reported that fewer caregivers were knowledgeable on the advantages of exclusive breastfeeding [29]. However, Assefa et al. [52] discovered that a higher percentage of women (85.6%) were aware that infants should only be breastfed for the first six months of their lives. The global exclusive breastfeeding stands at 44% [21]. Exclusive breastfeeding is a foundational element of the infant’s survival and child health because it is the most essential and irreplaceable food for a child’s growth and development. The fact that 40.7% of participants in this study are not aware of exclusive breastfeeding is a public health concern which calls for strengthening of exclusive breastfeeding education. However, a limitation is that this study did not ask the individuals, who were unaware of the six-month period of exclusive breastfeeding, about how long they believed children should be given only breastmilk.

The majority o’ respondents in this study (90.7%) agreed that breastfeeding offers prevention against diseases. According to results from other researches, breastfeeding is the best source of nutrition and disease prevention [40,53]. Studies show that breastfeeding protects children against a range of acute respiratory and gastrointestinal diseases until the age of six months, with the time of exclusive breastfeeding being marginally protective against otitis media even after breastfeeding has stopped [13,54,55]. Moreover, the risks of infectious diseases, such as gastrointestinal infections, otitis media, other respiratory tract infections and infection-induced wheezing may be reduced for several years after the termination of breastfeeding [56]. Spreading information about the additional benefits of breastmilk is therefore essential, with a focus on exclusive breastfeeding. According to this study’s findings, 82.6% of participants agreed that semi-solid foods should be introduced at 6 months and that breastfeeding should not be stopped when these foods are consumed. Similar to this, Assefa et al. [52] indicated that mothers were aware that complementary meals should not be offered to children until they are six months old. Moreover, mothers and caregivers were aware of the appropriate age to start complementary feeding [56]. However, their level of knowledge may impact how caregivers feed their children, hence it is recommended to strengthen infant and young child feeding education to a wider range of the public, including families and communities.

## 5. Conclusions

Caregivers of children between the ages of 0 and 24 months are aware of the advantages of breastfeeding and that it should continue even after the introduction of complementary feeding. They also understand that mothers who have HIV should breastfeed their infants. Caregivers lacked knowledge regarding exclusive breastfeeding, therefore there is a need to improve their education. The knowledge levels of caregivers were positively correlated with their educational backgrounds. Programmes to strengthen infant and young child feeding education should be intertwined with improving educational and health literacy.

## 6. Limitations

The exclusive breastfeeding and complementary feeding patterns were not examined in this study. The data were collected from a small sample; therefore, findings of this study cannot be generalised to the people residing in the Polokwane municipality. Funding was also another factor limited data collection in both rural and urban settings of the Polokwane municipality.

## Figures and Tables

**Figure 1 healthcare-11-01044-f001:**
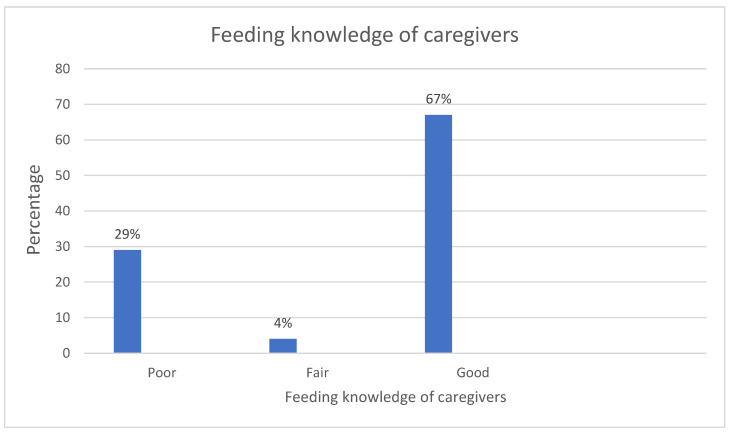
Infant and young child feeding knowledge of caregivers (on a scale of poor (0–50%), fair (51–80%) and good (81–100%).

**Table 1 healthcare-11-01044-t001:** Sociodemographic data of participants, % in rows (*n* = 86).

Variables	Categories	*n* (%)
Age of mother/caregiver	18–35 years	69 (80.2%)
	≥36 years	17 (19.8%)
Race	Black	85 (98.8%)
	Coloured	1 (1.2%)
Education status	Secondary education or less	48 (55.8%)
	Tertiary education	38 (44.2%)
Employment status	Temporary employed	7 (8.1%)
	Permanently employed	9 (10.5%)
	Self employed	11 (12.8%)
	Unemployed	59 (68.6%)
Source of income	Social grant	58 (67.4%)
	Pension fund	1 (1.2%)
	Salary	17 (19.8%)
	Other	10 (11.6%)
Number of household members	1–5	55 (63.9%)
	6–10	31 (36.0%)
Relationship to the child	Mother	84 (97.7%)
	Caregiver (grandmother, siblings of mother, nanny)	2 (2.3%)
Number of children	1–2	52 (60.5%)
	3–8	34 (39.5%)
Age of child in months	0–6	45 (52.3%)
	7–12	22 (25.6%)
	13–18	12 (13.9%)
	19–24	7 (8.1%)

**Table 2 healthcare-11-01044-t002:** Knowledge of participants regarding infant feeding, % in rows (*n* = 86).

Knowledge of Caregivers Regarding Infant Feeding	Yes	Not Sure	No
Should breastfeeding be initiated immediately after birth?	83 (96.5%)	2 (2.3%)	1 (1.2%)
Is exclusive breastfeeding up to 6 months?	51 (59.3%)	0 (0%)	35 (40.7%)
Does breastmilk protect the child from diseases?	78 (90.7%)	2 (2.3%)	6 (7%)
Can an HIV positive mother breastfeed?	56 (65.1%)	6 (7%)	24 (27.9%)
Should solid food be introduced at 6 months?	71 (82.6%)	4 (4.7%)	11 (12.7%)
Should breastfeeding be stopped when semi-solid and solid food are given?	11 (12.7%)	4 (4.7%)	71 (82.6%)

**Table 3 healthcare-11-01044-t003:** Knowledge of participants by sociodemographic profile % in rows (*n* = 86).

Knowledge of Family by Socio-Demographic Profile		Overall Knowledge			*p*-Values *
		Poor Knowledge	Fair Knowledge	Good Knowledge	
Age	18–35	3	20	46	0.751
	≥36	0	8	9	
Education	Secondary or less	1	3	0	<0.001
	Tertiary or more	2	25	55	
Employment	Employed	1	12	14	0.564
	Unemployed	2	16	41	
Number of children	1–2	2	16	34	0.436
	3–8	1	12	21	

* signifies statistical significance @ 95% CI.

## Data Availability

This article depends on the data gathered from caregivers of children aged 0–24 months in Seshego located in Polokwane municipality in Limpopo province of South Africa. The data generated or analysed during the current study is not openly accessible. However, it can be requested from corresponding author.

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
