# Peer review of "Infant and Young Child Feeding Knowledge among Caregivers of Children Aged between 0 and 24 Months in Seshego Township, Limpopo Province, South Africa"

_healthcare, 2023, doi:10.3390/healthcare11071044_

Round 1
Reviewer 1 Report
Reviewer note
Title: Infant and young child feeding knowledge among caregivers of children aged 0 to 24 months in Seshego Township, Limpopo province, South Africa
General comment:
- This manuscript presents the findings of and IYCF knowledge survey conducted in convenient samples of mothers with young children in S. Africa. While the information may have some relevance to inform local programs, the reviewer found it very narrow in scope. Authors could have used all the core and optional indicators of the WHO IYCF indicators to obtain a more comprehensive picture of the issue instead of just focusing of caregivers’ knowledge…Authors say it was a convenient sampling but applied a formula for representative cross-sectional study. It is difficult to tell if the findings are generalizable to all caregivers in the region (external validity?) Authors also leave important details in the methods section which makes it difficult for the reader to make sense of the finding. With careful revision, the paper may still have some merit for consideration.
See specific comments below.
Abstract:
- Lines 13-14: “Hence this study aims to determine IYCF knowledge of caregivers of children under 24 months.” Please include the setting. IYCF knowledge of caregivers living where/what setting (which country, rural or urban, etc.)
- Line 15 & 17: How does convenient sampling and inferential statistics go together? Were your samples representatives of the target population?
- Line 19: what the measure for “good IYCF” knowledge? Why not directly state what they knew about IYCF?
Background
- Lines 52-53: Since you cite an IYCF study from Ethiopia... authors could also add the following two IYCF studies in Ethiopia that are directly relevant to the aim of this paper:
[1. Ersino G, Henry CJ, Zello GA. Suboptimal Feeding Practices and High Levels of Undernutrition Among Infants and Young Children in the Rural Communities of Halaba and Zeway, Ethiopia. Food Nutr Bull. 2016 Sep;37(3):409-424. doi: 10.1177/0379572116658371. Epub 2016 Jul 9. PMID: 27402640.]
[2. Tessema M, Belachew T, Ersino G. Feeding patterns and stunting during early childhood in rural communities of Sidama, South Ethiopia. Pan Afr Med J. 2013 Feb 26;14:75. doi: 10.11604/pamj.2013.14.75.1630. PMID: 23646211; PMCID: PMC3641921.]
- Lines 71-79: This last paragraph provides lots of information but does not clearly indicate the gap in knowledge that triggered the current study. Why conduct this study if so much was known? What were the specific reason that warrant the current study in the selected locality?
-
Methodology
- Merge some of the sub-headings in this section
- Line 86: “The clinic” – This is the first mention of the clinic, but you refer to it as though the reader was already introduced to it earlier. Please describe the study setting well. Clearly indicate that your study was conducted in a healthcare facility setting…
- Lines 92-102: Where the final sample size? Also, why apply a sample size calculation formula for representative sample for a cross-sectional study although you indicated in the abstract that this was convenient sampling?
- What was your P (the population proportion) in the sample size calculation?
- Lines 105- 115: Describe your data collection instrument well. What were the key variables asked? What items did you use to measure the IYCF knowledge and practice of caregivers? What knowledge and demographic characteristics did you ask in the questionnaire?
Results
- Figure 1 presents feeding knowledge of caregivers (on a scale of poor, fair and good). However, this variable was not described in the methods section. What constitutes caregivers’ feeding knowledge? How was the scoring done for this variable?
- Table 3 – presents on overall knowledge. Again, what does overall knowledge refers to? How was it calculated? All this need to described in the method section
Discussion
- The study did not address IYCF practices, but the authors make references to IYCF practice.
Conclusion
- Limit your conclusion and recommendation to the key findings of your study.
Author Response
Reviewer comment: Authors say it was a convenient sampling but applied a formula for representative cross-sectional study. It was an error simple random sampling was used in this study
Reviewer comment: Authors also leave important details in the methods section which makes it difficult for the reader to make sense of the finding. Methodology has been expanded and more information added.
Reviewer comment: Lines 13-14 in abstract: “Hence this study aims to determine IYCF knowledge of caregivers of children under 24 months.” Please include the setting. IYCF knowledge of caregivers living where/what setting (which country, rural or urban, etc.) Setting included
Reviewer comment: Line 15 & 17 in abstract: How does convenient sampling and inferential statistics go together? Were your samples representatives of the target population?. Simple random sampling is correct method used in this study
Review comment: Line 19 in abstract: what the measure for “good IYCF” knowledge? Why not directly state what they knew about IYCF? Done as suggested
Review comment: Lines 52-53: Since you cite an IYCF study from Ethiopia... authors could also add the following two IYCF studies in Ethiopia that are directly relevant to the aim of this paper: [1. Ersino G, Henry CJ, Zello GA. Suboptimal Feeding Practices and High Levels of Undernutrition Among Infants and Young Children in the Rural Communities of Halaba and Zeway, Ethiopia. Food Nutr Bull. 2016 Sep;37(3):409-424. doi: 10.1177/0379572116658371. Epub 2016 Jul 9. PMID: 27402640.] [2. Tessema M, Belachew T, Ersino G. Feeding patterns and stunting during early childhood in rural communities of Sidama, South Ethiopia. Pan Afr Med J. 2013 Feb 26;14:75. doi: 10.11604/pamj.2013.14.75.1630. PMID: 23646211; PMCID: PMC3641921.] The references were cited and included
Review comment: Lines 71-79: This last paragraph provides lots of information but does not clearly indicate the gap in knowledge that triggered the current study. Why conduct this study if so much was known? What were the specific reason that warrant the current study in the selected locality? The reasons why this study was conducted were stated
Review comment: Merge some of sub-headings in methodology. Research design merged with setting
Review comment: Line 86: “The clinic” – This is the first mention of the clinic, but you refer to it as though the reader was already introduced to it earlier. Please describe the study setting well. Clearly indicate that your study was conducted in a healthcare facility setting… The description of study design has been revised and enriched.
Review comment: Lines 92-102: Where the final sample size? Also, why apply a sample size calculation formula for representative sample for a cross-sectional study although you indicated in the abstract that this was convenient sampling? Simple random sampling was used instead
Review comment: What was your P (the population proportion) in the sample size calculation? P described Review comment: Lines 105- 115: Describe your data collection instrument well. What were the key variables asked? What items did you use to measure the IYCF knowledge and practice of caregivers? What knowledge and demographic characteristics did you ask in the questionnaire? Data collection section has been revised and enriched
Review comment: Figure 1 presents feeding knowledge of caregivers (on a scale of poor, fair and good). However, this variable was not described in the methods section. What constitutes caregivers’ feeding knowledge? How was the scoring done for this variable? The scaling of knowledge has been described in the methodology
Review comment: The study did not address IYCF practices, but the authors make references to IYCF practice. The point on practices has been revised Review comment: Limit your conclusion and recommendation to the key findings of your study. Done
Reviewer 2 Report
Thank you for the opportunity to review your manuscript. It is certainly of interest to your local community and adds to the knowledge of maternal education and feeding practices in your region. It is of value to have sound research from communities in the majority world. How does this work contribute to the international literature? Can you comment on the suitability of your survey to capture relevant, reliable information that perhaps others across the world might want to discover in their own community?
In general, the English writing is fine, however there are some spelling mistakes to be corrected. There is inconsistent use of terminology e.g. children/ kids.
Introduction.
line 43 - lower than what?
line 45 - it would be useful to say which standards you are referring to.
Line 49 - I would remove the word 'bad' here as it implies the children are making bad decisions, which is incorrect.
Line 68 - "educated/ informed' women were 5.5 times... there is a word missing here.
Line 77 - do you mean to 'blame' the caregivers? Is it their fault? or are there wider systemic reasons why they don't have good feeding practices?
Why is it useful to determine the knowledge of this particular region? Are there indications that there are feeding problems in this region? or that mothers are poorly informed or that existing findings cannot be generalised to this population? there could be an indication of the future consequences of knowing this information - will there be a future project to address a shortfall if that becomes clear?
Method -
Research Design should include 'survey'. Sample size calculator should also indicate the size of the sample required, as well as the formula.
Over what time period was the data collected? Where there any considerations of the participants literacy skills? What is the 'existing literature' that this survey is based on? If there is an existing survey, it should be mentioned in the introduction as the basis for this study.
Please can you clarify the scoring e.g. there are 6 questions, each question is therefore worth 16%. How was a 'not sure' response scored? Were the scores added up out of percentages, or were they added up out of 6 and then converted to a percentage e.g. 3 out of 6 or better is considered to be 'good'? This would be equivalent to a 50:50 chance.
Table 3. For each profile, the sample size needs to be included.
The education levels in Table 3 are not the same a those detailed in Table 1. It would be useful to include 'primary or less' in Table 1.
Discussions
There needs to be some discussion of test parametrics before claiming that a correct response to 3 out of 6 questions is 'sufficient understanding' of infant and child feeding. Can you say that there is sufficient understanding, if only 60% of mothers know about exclusive breastfeeding? Perhaps some of the questions are more important than others.
The discussion seems very binary. The mother must breastfeed for 6 months, and then introduce solids at 6 months and continue breastfeeding. There is no nuance or consideration of a range of feeding practices e.g. 5 and a half months to 6 and a half months. I wonder how much the wording of the questions affects the responses. 60% of participants knew about exclusive breastfeeding upto 6 months, but 82% knew to introduce solids at 6 months - does this mean that they saw 6 months as the latest time to introduce solids? or that there is a mixed understanding of this? Some of this is addressed in the limitations and I would move this section up into the discussion as it is relevant there.
Conclusion is sound.
Author Response
Review comment: Can you comment on the suitability of your survey to capture relevant, reliable information that perhaps others across the world might want to discover in their own community? Information added
Review comment: In general, the English writing is fine, however there are some spelling mistakes to be corrected. There is inconsistent use of terminology e.g. children/ kids. Children maintained.
Review comment: line 43 - lower than what? The sentence was extended. Review comment: line 45 - it would be useful to say which standards you are referring to. The standards referred to described.
Review comment: Line 49 - I would remove the word 'bad' here as it implies the children are making bad decisions, which is incorrect. Removed.
Review comment: Line 68 - "educated/ informed' women were 5.5 times... there is a word missing here. It has been clarified and expended.
Review comment: Line 77 - do you mean to 'blame' the caregivers? Is it their fault? or are there wider systemic reasons why they don't have good feeding practices? Statement has been revised.
Review comment: Why is it useful to determine the knowledge of this particular region? Are there indications that there are feeding problems in this region? or that mothers are poorly informed or that existing findings cannot be generalised to this population? there could be an indication of the future consequences of knowing this information - will there be a future project to address a shortfall if that becomes clear? The reasons for conducting this study was outlined.
Reviiew comment: Research Design should include 'survey'. Sample size calculator should also indicate the size of the sample required, as well as the formula. Included as suggested
Review comment: Over what time period was the data collected? Where there any considerations of the participants literacy skills? What is the 'existing literature' that this survey is based on? If there is an existing survey, it should be mentioned in the introduction as the basis for this study. Period stated and the questionnaire was based on WHO literature.
Review comment: Please can you clarify the scoring e.g. there are 6 questions, each question is therefore worth 16%. How was a 'not sure' response scored? Were the scores added up out of percentages, or were they added up out of 6 and then converted to a percentage e.g. 3 out of 6 or better is considered to be 'good'? This would be equivalent to a 50:50 chance. It has been clarified.
Review comment: Table 3. For each profile, the sample size needs to be included. It has been included.
Review comment: The education levels in Table 3 are not the same a those detailed in Table 1. It would be useful to include 'primary or less' in Table 1. Error corrected.
Review comment: The discussion seems very binary. The mother must breastfeed for 6 months, and then introduce solids at 6 months and continue breastfeeding. There is no nuance or consideration of a range of feeding practices e.g. 5 and a half months to 6 and a half months. I wonder how much the wording of the questions affects the responses. 60% of participants knew about exclusive breastfeeding upto 6 months, but 82% knew to introduce solids at 6 months - does this mean that they saw 6 months as the latest time to introduce solids? or that there is a mixed understanding of this? Some of this is addressed in the limitations and I would move this section up into the discussion as it is relevant there. Revised and some limitations were moved up in the discussion section.
Review comment: Conclusion is sound. Maintained
Reviewer 3 Report
Although assessing caregivers’ knowledge about the feeding of infants and young children is an important topic to study, the manuscript of Muleka et al should be revised to be acceptable for publication.
Abstract: L9-13 the background information is very long while minimal information was presented in the Methods section about the study design and specifically about the questionnaire which was used to assess the knowledge of the participants.
L19- Good knowledge should be defined
L25-26- unclear: who is the target? “Educational and health literacy” is very general; please be specific
Introduction: The relevance of the study for the community studied should be added. Is feeding infants and young children a concern in Seshego Township in Limpopo province, or in South Africa?
Methodology:
L82: Quantitative study is unclear. When was the study conducted?
Provide more information about the setting, for example, how many 0-24 children the clinic serves per year, month, or day?
L97-102: What was the calculated sample size?
L109: Give the references of the literature mentioned
L110: “did not feature” should be re-worded
L111: Supervisors – please specify who are the supervisors. How many “supervisors” and dietitians were involved in the content validation of the questionnaire?
Results:
Fig. 1: Add a footnote indicating the score ranges for Good, Fair, and Poor child-feeding knowledge
Discussion:
L 176-183: Is the repetition of the Results – should be removed
Conclusions: L247-250: the sentence should be re-phrased for clarity
The authors mentioned that caregivers with low educational levels had the least child-feeding knowledge. This is not surprising. However, the authors should make specific recommendations for an effective way to improve the child-feeding knowledge of caregivers with low educational levels.
Author Response
Review comment: Abstract: L9-13 the background information is very long while minimal information was presented in the Methods section about the study design and specifically about the questionnaire which was used to assess the knowledge of the participants. More information on questionnaire added.
Review comment: L19- Good knowledge should be defined. It was defined.
Review comment: L25-26- unclear: who is the target? “Educational and health literacy” is very general; please be specific. Done
Review comment: Introduction: The relevance of the study for the community studied should be added. Is feeding infants and young children a concern in Seshego Township in Limpopo province, or in South Africa? It was outlined and added.
Review comment: L82: Quantitative study is unclear. When was the study conducted? Provide more information about the setting, for example, how many 0-24 children the clinic serves per year, month, or day? It has been indicated and information on the setting provided.
Review comment: L97-102: What was the calculated sample size? Indicated.
Review comment: L109: Give the references of the literature mentioned. Done.
Review comment: L110: “did not feature” should be re-worded. Done.
Review comment: L111: Supervisors – please specify who are the supervisors. How many “supervisors” and dietitians were involved in the content validation of the questionnaire? Done.
Review comment: Fig. 1: Add a footnote indicating the score ranges for Good, Fair, and Poor child-feeding knowledge. It was added.
Review comment: Discussion: L 176-183: Is the repetition of the Results – should be removed. Removed.
Review comment: Conclusions: L247-250: the sentence should be re-phrased for clarity. Done. The authors mentioned that caregivers with low educational levels had the least child-feeding knowledge. This is not surprising. However, the authors should make specific recommendations for an effective way to improve the child-feeding knowledge of caregivers with low educational levels. Clarified in the discussion section.
Round 2
Reviewer 1 Report
No further comment.
Author Response
Thank you
Reviewer 2 Report
The changes made have greatly enhanced this manuscript.
My main concern now is that the introduction is quite dense with long paragraphs that do not adequately provide a rationale for the study - why this study in this location? The most interesting part of the introduction, lines 89-92, regarding relevant research in Limpopo region, is lost, whereas it in fact forms a sound basis for conducting this research. The local research that forms the basis of this study should be emphasised in a paragraph of its' own with more detail. Less space could be given to the known health benefits of breastfeeeding to accomodate this. This should be mirrored in the abstract.
Author Response
The rationale for conducting the study has been clearly outlined in the background and further in the background. longer paragraphs shortened
Reviewer 3 Report
Authors should indicate the definition of "good" knowledge by adding the score range considered "good" in the abstract. The rest of the previous comments were addressed by the authors. Further language editing is required.
Author Response
a score for the good knowledge added in the abstract and editing done